# Oxygenated organic molecules produced by low-NO$_x$ photooxidation of aromatic compounds: contributions to secondary organic aerosol and steric hindrance

Xi Cheng[1], Yong Jie Li[2], Yan Zheng[1], Keren Liao[1], Theodore K. Koenig[1], Yanli Ge[1], Tong Zhu[1],
5   Chunxiang Ye[1], Xinghua Qiu[1], Qi Chen[1]

[1]State Key Joint Laboratory of Environmental Simulation and Pollution Control, BIC-ESAT and IJRC, College of Environmental Sciences and Engineering, Peking University, Beijing, China

[2]Department of Civil and Environmental Engineering, Department of Ocean Science and Technology, and Centre for Regional Oceans, Faculty of Science and Technology, University of Macau, Taipa, Macau, China

10   **Correspondence:** Qi Chen (qichenpku@pku.edu.cn) and Yong Jie Li (yongjieli@um.edu.mo)

**Abstract.** Oxygenated organic molecules (OOMs) produced by the oxidation of aromatic compounds are key components of secondary organic aerosol (SOA) in urban environments. The steric effects of substitutions and rings and the role of key reaction pathways on altering the OOM distributions remain unclear because of the lack of systematic multi-precursor study over a wide range of OH exposure. In this study, we conducted flow-tube experiments and used the nitrate adduct time-of-flight chemical ionization mass spectrometer ($NO_3^-$-TOF-CIMS) to measure the OOMs produced by the photooxidation of six key aromatic precursors under low-$NO_x$ conditions. For single aromatic precursors, the detected OOM peak clusters show one or two oxygen-atom difference, indicating the involvement of multi-step auto-oxidation and alkoxy radical pathways. Multi-generation OH oxidation is needed to explain the diverse hydrogen numbers in the observed formulae. Especially for double-ring precursors at higher OH exposure, multi-generation OH oxidation may have significantly enriched the dimer formulae. The results suggest that methyl substitutions in precursor lead to less fragmented OOM products, while the double-ring structure corresponds to less efficient formation of closed-shell monomeric and dimeric products, both highlighting significant steric effects of precursor molecular structure on the OOM formation. Naphthalene-derived OOMs however have lower volatilities and greater SOA contributions than the other-type of OOMs, which may be more important in initial particle growth. Overall, the OOMs identified by the $NO_3^-$-TOF-CIMS may have contributed up to 30.0% of the measured SOA mass, suggesting significant mass contributions of less oxygenated, undetected semi-volatile products. Our results highlight the key roles of progressive OH oxidation, methyl substitution and ring structure in the OOM formation from aromatic precursors, which needs to be considered in future model developments to improve the model performance on organic aerosol.

## 1 Introduction

Oxidation of volatile organic compounds (VOCs) leads to the formation of ozone and secondary organic aerosols (SOA) (Ziemann and Atkinson, 2012). Light and heavy aromatic VOCs are important SOA precursors that can be emitted from various anthropogenic sources such as transportation, solvent use, wood burning, coal burning, and cooking (Henze et al., 2008; Pye and Pouliot, 2012). There are many chamber studies on the SOA formation from aromatic VOCs (Li et al., 2016a; Li et al., 2016b; Lambe et al., 2011; Ng et al., 2007; Chan et al., 2009; Kautzman et al., 2010). The derived SOA yields vary by precursors and $NO_x$ levels owing to the difference in forming condensable products, i.e., oxygenated organic molecules (OOMs). Characterization of these OOM species is however limited and challenging. Most of atmospheric chemical transport models (CTM) still use SOA-yield-based parameterizations, which have difficulties in reproducing the concentration and the variability of organic aerosol in urban areas (Tsigaridis et al., 2014).

Recent developments in time-of-flight chemical ionization mass spectrometry provide new insights into the formation of OOMs and their contribution to SOA (Bianchi et al., 2019). Wang et al. (2017) indicate that intramolecular H-shift of peroxy radicals ($RO_2$) from alkyl benzenes can compete with bimolecular reactions under atmospheric conditions, thereby substantially increasing the product oxygen numbers via auto-oxidation. Laboratory studies show further evidence for the occurrence of both of multi-generation OH oxidation and multi-step auto-oxidation in the aromatic oxidation to increase the oxidation state of the products (Garmash et al., 2020; Cheng et al., 2021b). Additionally, Berndt et al. (2018b) suggest high formation rates of accretion products from self- and cross-reactions of $RO_2$ radicals for aromatic precursors, resulting in great carbon and oxygen number increments that lower the product volatility. The three mechanisms, i.e., multi-generation OH oxidation, multi-step auto-oxidation, and accretion reactions, play different roles under different oxidation conditions. Moreover, aromatic precursors differ in substituents and ring numbers, which may cause steric effects in the OOM formation. Cheng et al. (2021b) suggest that for benzene and toluene, the multi-generation OH oxidation likely proceeds more favorably by H abstraction than OH addition and the dimer formation is unfavorable at high OH exposures. Other laboratory studies indicate that under similar OH exposures, the oxidation of isopropylbenzene, ethylbenzene and toluene likely forms more oxidized $RO_2$ radicals than the oxidation of benzene does (Wang et al., 2017). The formation of dimeric products from the oxidation of trimethylbenzene seemed being more significant for meta-substituents than for ortho-substituents (Wang et al., 2020b). However, there is still a lack of multi-precursor studies that compares the OOM formation systematically under a wide range of OH exposure. A better understanding of the precursor steric effects and the OOM potential of forming SOA is needed to assist future developments of detailed model representation on the SOA formation.

Herein, the formation of OOMs from six aromatic VOCs (i.e., benzene, toluene, m-xylene, 1,3,5-trimethylbenzene, naphthalene, and 1-methylnaphthalene) is investigated in the flow-tube experiments under low-$NO_x$ conditions, aiming at gaining insights into the OH-initiated chemistry and the low-$NO_x$ representation of suburban and downwind environments. The OOMs are characterized by using a nitrate adduct time-of-flight chemical ionization mass spectrometer ($NO_3^-$-TOF-CIMS). The experiments cover a wide range of OH exposures that are equivalent to approximately 1 - 19 days assuming a mean OH

concentration of $1.5 \times 10^6$ molecules cm$^{-3}$. The influences of precursor structure and OH exposure on the formation of OOMs and the contributions of the identified OOMs to SOA are investigated.

## 65 2 Methods

### 2.1 Experimental procedures

Experiments were conducted in a 13.3-L Aerodyne oxidation flow reactor (OFR) with a mean residence time of 95 s under low-NO$_x$ conditions. The operation of the OFR has been described previously by Cheng et al. (2021b). The OFR was operated in an OFR254-5 mode, with 254 nm lights on and 5 ppm of externally generated O$_3$ introduced into the OFR (Lambe et al., 70 2017; Peng et al., 2018). The OH radicals are generated via:

$$O_3 + hv \ (254 \ nm) \rightarrow O_2 + O(^1D) \tag{R1}$$

$$O(^1D) + H_2O \rightarrow 2OH \tag{R2}$$

The relative humidity (RH) inside the OFR was maintained at 23.4 - 29.4%. Concentrations of OH radicals were varied by ramping the voltage of the UV lamps in the OFR, for which each lamp voltage represents one experimental condition. All 75 experiments herein were conducted without injection of seed particles and NO$_x$. We estimate a background NO$_x$ level of < 1 ppb throughout the experiments. In total, 27 experiments were performed for the six aromatic precursors. Table 1 lists the experimental conditions and the key measured and derived quantities, including those for the oxidation of benzene and toluene that have been discussed in our previous study (Cheng et al., 2021b). The concentrations of OH and HO$_2$ radicals were estimated by using a photochemical box model (PAMchem) (Lambe et al., 2017; Cheng et al., 2021b). The results suggest that 80 the first step of oxidation of the aromatic VOCs was dominated by the OH radicals rather than by O$_3$ because of much higher reaction rate constants for the former than the latter (Lambe et al., 2017; Peng et al., 2016). The reaction rates of early-generation oxidation products that contain double bounds with O$_3$ are also likely slower than those with OH radicals (Molteni et al., 2018; Wang et al., 2020b). Therefore, we believe that the oxidation chemistry was prevalently initiated by OH radicals in our experiments. The modeled HO$_2$-to-OH concentration ratio is about 2-18, while the rate constant for RO$_2$ + HO$_2$ is 85 typically an order of magnitude smaller than that for RO$_2$ + OH (Peng and Jimenez, 2020, and references therein). Figure S1 shows the calculated contributions of various pathways to the RO$_2$ loss in our experiments for a given set of rate constants. Both of the RO$_2$ + HO$_2$ and RO$_2$ + OH channels are important for RO$_2$ loss under conditions of our experiments considering the uncertainties of the estimations and the variations of reaction constants. The RO$_2$ isomerization rate coefficients are highly structure dependent, ranging from ~10$^{-3}$ to a few tens of s$^{-1}$ for aromatic RO$_2$ (Wang et al., 2017; Praske et al., 2018). Therefore, 90 the fast RO$_2$ + HO$_2$ reactions may limit some RO$_2$ auto-oxidation in our experiments.

### 2.2 Measurements

The concentration of $O_3$ was monitored with an UV photometric analyzer (2B Technologies, 202). Concentrations of aromatic VOCs were measured by an IONICON proton transfer reaction-quadrupole interface time-of-flight mass spectrometer (PTR-QiTOF). Calibration of PTR-QiToF was conducted by using gas standards (Spectra Gases, ~1 ppm) at different concentration levels. The measurements uncertainty is about 15% for calibrated species (Huang et al., 2019). Particle size distributions were measured by a scanning mobility particle sizer (SMPS; TSI, 3938). The SOA mass concentrations were measured by an Aerodyne long time-of-flight soot particle aerosol mass spectrometer (LTOF-SP-AMS). Calibrations of ionization efficiency (IE) and relative IE (RIE) followed the standard procedures by using pure ammonium nitrate ($NH_4NO_3$) and ammonium sulfate (($NH_4)_2SO_4$) (Zheng et al., 2020). A collection efficiency (CE) value of 1 was applied to the LTOF-SP-AMS data herein, which leads to a good agreement between the AMS and SMPS results. The oxygen-to-carbon (O:C) and hydrogen-to-carbon (H:C) ratios of SOA were estimated by the method introduced by Aiken et al. (2008). Because the measured ratios of $(H_2O^+)_{org}/(CO_2^+)_{org}$ (i.e., around 0.33) and $(CO^+)_{org}/(CO_2^+)_{org}$ (i.e., 1.1-1.2) for chamber aromatic SOA were similar to the default ratios of 0.225 and 1, respectively (Chhabra et al., 2011; Nakao et al., 2013), we did not update these ratios in the calculation of elemental ratios (Chen et al., 2011).

The OOMs were measured by the Aerodyne $NO_3^-$-TOF-CIMS in the form of deprotonated ions, clusters with a nitrate monomer, or clusters with a nitrate dimer. This study only presents clusters having one nitrate ion. The oxidation of benzene and toluene have been discussed in our previous study for a fitted mass-to-charge ratio ($m/z$) range of 150 to 450 Thomson (Th) (Cheng et al., 2021b). In this study, we refitted the spectra to up to 650 Th, resulting in slightly different molar yields (Table S1). Although our work and other studies report similar calibration factors for the range of 100-450 Th for high-resolution time-of-flight (HTOF) type of $NO_3^-$-CIMS (Ehn et al., 2014; Jokinen et al., 2012; Kürten et al., 2012), it does not guarantee a flat transmission on a wider range of $m/z$. The transmission correction is instrument-specific and strongly depends on the chosen settings of the instrument (Heinritzi et al., 2016). Tian et al. (2023) and Zheng et al. (2023) show a relatively smooth transmission curve for $m/z < 450$ Th with an enhance factor of 1-1.2. We used a uniform calibration factor of $1.23 \times 10^{10}$ molecules cm$^{-3}$ with additional sampling-line vapor loss correction of 26% for all OOMs in this study, with considerations of the transmission uncertainty in the result interpretation. This may underestimate the sensitivity of large molecules by a factor of 2 if we consider an average transmission efficiency of ~0.5 for 450-650 Th suggested by Tian et al. (2023). The background signals of individual OOMs were determined from measurements made without the injection of VOC precursors. The instrument operation and data analysis have been described in detail in our previous studies (Cheng et al., 2021a; Cheng et al., 2021b).

**2.3 Dimer formation rates**

Dimeric products may be formed by the accretion reactions between $RO_2$ and $R´O_2$ radicals. At steady state, the formation rate of dimers is represented as follows:

$$\frac{d[ROOR´]}{dt} = \sum_i^n k_{R,i}[RO_2][R´O_2] - k_{loss}[ROOR´] = 0 \qquad (1)$$

where $k_{R,i}$ (cm$^3$ molecule$^{-1}$ s$^{-1}$) is the apparent rate constants of accretion reactions ($i \geq 1$), and $k_{loss}$ (s$^{-1}$) represents the loss rate of OOMs (0.04-0.1 s$^{-1}$) (Section S1 of the Supplement). We use Eq. 1 to provide rough constraints to the formation rate constants of detected dimeric products, which is similar to the approach used by Molteni et al. (2019). There are 46, 54, and 76 assigned dimeric products from benzene, toluene, and naphthalene experiments, respectively, from the NO$_3^-$-TOF-CIMS measurements of OOM molecular formulae. However, for each precursor, only 5-8 dimers have sufficient combinations of above-detection-limit signals of possible RO$_2$ and R´O$_2$ to solve $k_{R,i}$ by using a non-negative multi-linear regression method (Sklearn from Python 3). Only the $k_{R,i}$ values that are greater than zero but less than $9\times10^{-10}$ cm$^3$ molecule$^{-1}$ s$^{-1}$ (considering the collision limit) with correlation coefficients ($r$) of $> 0.5$ are considered as reasonable rate constants and are listed in Table 2. Theoretical and laboratory studies indicate that the production rate constant of ROOR´ can reach $10^{-10}$ cm$^3$ molecules$^{-1}$ s$^{-1}$ (Molteni et al., 2019; Hasan et al., 2020; Perakyla et al., 2023). The OOMs that form slowly would not be detectable by the NO$_3^-$-TOF-CIMS because of the fast vapor losses at the SOA loading level at the exit of OFR. The detected dimeric products (e.g., C$_{14}$H$_{18}$O$_{14}$) showed rapid increases to stable concentrations with time series similar with their parent RO$_2$ (e.g., C$_7$H$_9$O$_{10}$), indicating a fast production of these dimers. The PAMchem model simulations indicate minor changes of the ROOR´ concentrations when the reaction rate constant of RO$_2$ + R´O$_2 \rightarrow$ ROOR´ reaches above $5\times10^{-11}$ cm$^3$ molecules$^{-1}$ s$^{-1}$ (Figure S2). We therefore think the steady-state assumption may stand for the few dimers that were listed in Tables S2-S7.

**2.4 Contribution of OOMs to SOA**

The contributions of the detected OOMs to SOA were only estimated for benzene, toluene, and naphthalene experiments, for which both LTOF-SP-AMS and SMPS measurements are available. We consider both of the irreversible and equilibrium-type condensation. The former was represented by the net condensation flux of the observed OOMs, which can be described by using an aerosol growth model (Tröstl et al., 2016). The latter was represented by the particulate fraction estimated by the partitioning theory and the vapor pressure of OOMs (Pankow, 1994). The calculation details are provided in Section S1 of the Supplement. Aromatic OOMs consist of abundant hydroxyl (-OH) and carboxylic (=O) groups but less hydroperoxide (-OOH) groups compared to monoterpene OOMs (Wang et al., 2020a). We therefore applied the empirical volatility parameterization that was used in previous naphthalene studies to estimate the saturation concentrations ($C^*$) of the detected OOMs (Epstein et al., 2010; Donahue et al., 2011). The OOMs were grouped into volatility bins based on the volatility-basis-set (VBS) method (Donahue et al., 2006). They were classified as ultralow volatility organic compounds (ULVOC, $\log C^* \leq -9.5$), extremely low volatility organic compounds (ELVOC, $-9.5 < \log C^* \leq -4.5$), low volatility organic compounds (LVOC, $-4.5 < \log C^* \leq -0.5$), semi-volatile organic compounds (SVOC, $-0.5 < \log C^* \leq 2.5$), and intermediate volatility organic compounds (IVOC, $2.5 < \log C^* \leq 6.5$) (Bianchi et al., 2019; Schervish and Donahue, 2020). In this study, SOA contributions from OOMs with $C^*$ of $\leq$ 0.01 μg m$^{-3}$ were calculated by the aerosol growth model, and those from OOMs with $C^* > 0.01$ μg m$^{-3}$ were calculated by the partitioning method.

## 3 Results and discussion

### 3.1 Product distributions

The general oxidation chemistry for OH-initiated pathways of aromatic VOCs is described in Section S2 of the Supplement. Figure 1 shows the OOM product distributions for the oxidation of the six aromatic precursors under similar OH exposures of approximately $(1.2\text{-}1.5)\times10^{12}$ molecules $cm^{-3}$ s. For a given precursor of $C_xH_y$, we name the detected OOMs as monomeric products ($C_{<x}$ and $C_x$ series) and dimeric products ($C_{<2x}$ and $C_{2x}$ series). Monomeric products contribute to the majority of the detected OOM signals for the six aromatic precursors studied herein, while the dimeric products contribute to up to 18% of the OOM signals. The highest dimeric signal fraction is observed for m-xylene OOMs. Overall, ion peaks in the $m/z$ range of 200-650 Th show in clusters with progressions and sequences that preserve the precursor carbon structure. The lower ends of the peak clusters of monomeric products for benzene, toluene, m-xylene, and 1,3,5-trimethylbenzene are shifted by progressive differences of 14 Th ($-CH_2$). The products from naphthalene and 1-methylnaphthalene also show a similar $-CH_2$ progression. The progression is doubled for dimeric products (28 Th, $-(CH_2)_2$). Moreover, there are series of clusters with peak sequences that differ by one and two oxygen atoms in both of monomeric and dimeric products, respectively, in each mass spectrum. The peak clusters then differ by hydrogen atom numbers for each oxygen atom addition. The general shifting of peak abundance to products with 1-4 more hydrogen (i.e., to the right $m/z$ side) is consistent with the importance of OH addition in the oxidation of aromatic precursors. By contrast, dimeric clusters show abundant distributions in the hydrogen-deficient $m/z$ values (i.e., on the left side), suggesting significant H abstraction process before the accretion reactions occur.

The $C_{<x}$ products are typically fragmented products formed from the decomposition of alkoxy (RO) radicals (Li and Wang, 2014). Methyl substitution may prevent the efficient formation of phenoxy radicals, thereby lowering the abundance of fragmented products (Schwantes et al., 2017). Indeed, we observed high signal fractions of $C_{<x}$ products (59.4-65.7%) in OOMs produced by the oxidation of benzene, toluene, naphthalene and 1-methylnaphthalene that have low methyl-to-aryl carbon ratios ($C_{Me}:C_{Ar} < 0.2$), whereas the signal fractions of $C_{<x}$ products produced by the oxidation of m-xylene and 1,3,5-trimethylbenzene ($C_{Me}:C_{Ar} > 0.2$) are much lower (17.9-26.3%). Tables S2-S7 in the Supplement list the formulae and the relative signals of major OOMs for the experiments presented in Figure 1. For the oxidation of benzene, toluene, naphthalene and 1-methylnaphthalene, $C_3H_4O_5$ and $C_4H_4O_5$ are the main common $C_{<x}$ products. These molecules are likely second- or third-generation products formed via the fragmentation of phenoxy radicals with rich unsaturated aryl carbon moieties (Schwantes et al., 2017). $C_5H_6O_6$ and $C_8H_{10}O_7$ are the most abundant $C_{<x}$ products for the oxidation of m-xylene and 1,3,5-trimethylbenzene, respectively. $C_5H_6O_6$ is likely a ring-scission product (e.g., carboxylic acid), while $C_8H_{10}O_7$ may be a fragmented ring-retaining product formed by the dealkylation pathway (Zaytsev et al., 2019; Mehra et al., 2020).

The $C_x$ products with odd H numbers in their formulae for single-precursor oxidation systems are plausibly open-shell products (i.e., mainly $RO_2$) (Zhao et al., 2018), and the $C_x$ products with even H numbers are normally closed-shell products. Central in the formation of OOMs from aromatic VOCs, the bicyclic peroxy radicals (BPRs) contain 5 oxygen numbers ($C_xH_{y+1}O_5$),

which are formed by two steps of $O_2$ addition after OH addition (Birdsall et al., 2010). As shown in Table S8, the signal fractions of $C_xH_{y+1}O_7$ and $C_xH_{y+1}O_9$ (0.1-1.0%) are greater than those of $C_xH_{y+1}O_5$ (0.05-0.4%), explained by aggressive multiple steps of auto-oxidation. These odd-oxygen $RO_2$ radicals are the results of the predominant OH-addition plus sequential

$O_2$ incorporation (auto-oxidation). Besides, even-oxygen $RO_2$ radicals are observed, with $C_xH_{y+1}O_6$ and $C_xH_{y+1}O_8$ (0.1-1.1%) being the most abundant. Their formation might involve the RO pathway (Orlando et al., 2003; Cheng et al., 2021b). In the RO pathway, the RO radicals undergo intermolecular H-shift (isomerization) instead of decomposition, which forms C-centered alkyl radicals that are ready for further auto-oxidation. For example, the BPR ($C_xH_{y+1}O_5$) can react with $HO_2$ to form the RO radical $C_xH_{y+1}O_4$. The newly formed RO radical then isomerizes to become a hydroxylated alkyl radical and results in

a new $RO_2$ radical ($C_xH_{y+1}O_6$) via $O_2$ addition. The most abundant closed-shell $C_x$ products are $C_6H_6O_6$ (3.1%), $C_7H_8O_6$ (3.8%), $C_8H_{10}O_7$ (8.6%), $C_9H_{14}O_7$ (17.5%), $C_{10}H_8O_6$ (2.9%), and $C_{11}H_{12}O_{10}$ (2.1%) for the oxidation of benzene, toluene, m-xylene, 1,3,5-trimethylbenzene, naphthalene and 1-methylnaphthalene, respectively. We obtained average O:C ratios of 0.9-1.1 for closed-shell OOM products formed from monocyclic precursors and 0.7-0.8 for those formed from double-ring precursors. The lower O:C ratios for the latter is probably because the second aromatic ring prohibits extensive auto-oxidation (Molteni

et al., 2018).

Among the dimeric products, the $C_{2x}$ products are much more abundant than the $C_{<2x}$ ones in terms of both number and amount (Tables S2-S7). The observed $C_{2x}$ dimers have odd and even oxygen numbers in their formulae, suggesting the occurrence of both of the cross- and self-reactions of $RO_2$ radicals. For example, $C_{18}H_{26}O_{12}$ is a product from the 1,3,5-trimethylbenzene oxidation, which may be formed via self-reactions of $C_9H_{13}O_7$ or cross-reactions of other even-oxygen $RO_2$ radicals. Dimers

that are formed from BPR or $RO_2$ with further auto-oxidation are expected to contain at least 8 oxygen atoms. Interestingly, the $C_{2x}$ products with 4-7 oxygen numbers are abundant, suggesting the occurrence of accretion reaction between BPR and less oxygenated $RO_2$ radicals. These $C_{2x}$ products show a greater abundance for the oxidation of m-xylene (Fig. 1c) than for those of other precursors, indicating more stable accretion reactions for the less oxygenated $RO_2$ radicals derived from m-xylene due to stereoselectivity. Consistently, Wang et al. (2020b) identified more abundant dimeric products for the meta-

substitution aromatic. Moreover, different numbers of hydrogen atoms in the dimer formulae suggest different formation routes. For monocyclic aromatic precursors, the main $C_{2x}$ products are $C_{2x}H_{2y+2}O_{10-14}$, likely formed by the accretion reactions of two $C_xH_{y+1}O_z$ radicals (Table S8). These $C_xH_{y+1}O_z$ are usually $RO_2$ radicals formed from multi-step auto-oxidation with one step of OH addition. Unlike the monocyclic precursors, the main $C_{2x}$ products produced by the oxidation of naphthalene and 1-methylnaphthalene are $C_{2x}H_{2y}O_{10-14}$, $C_{2x}H_{2y+2}O_{10-14}$, and $C_{2x}H_{2y+4}O_{10-14}$, showing more diverse H number distributions. This

result suggests more efficient multi-generation OH oxidation (e.g., adding one H atom via OH addition vs. losing one H atom via H abstraction) before accretion reactions occur for double-ring aromatic precursors than for single-ring precursors.

The apparent molar yields of the OOMs detected by the $NO_3^-$-TOF-CIMS at the exit of OFR were calculated as $(k_{loss} \times [OOMs])/(k_1 \times [VOC] \times [OH])$, where $k_{loss}$ is the loss rate constant of OOMs and $k_1$ is the VOC-OH reaction rate constant. The AMS and SMPS particle measurements are only available for benzene, toluene, and naphthalene, for which we can correct

the vapor loss of OOMs to the particle phase and the OFR wall. Figure 2 shows the OOM yields under different OH exposures. The yields range from 0.07% to 2.05%, among which the yields of naphthalene-derived OOMs are greater than those of the other two monocyclic aromatic precursors. This is consistent with the result from Molteni et al. (2018), which obtained a higher OOM yield for naphthalene (1.8%) than for benzene (0.2%) and toluene (0.1%). As the OH exposure increases, the OOM yields for benzene and toluene first increase and then decrease after the equivalent photochemical age exceeds 7 days.

The initial increase of OOM yields is consistent with the results presented by Garmash et al. (2020) for the oxidation of benzene under low OH exposure. The later decrease of the apparent OOM yields under high OH exposure suggest the production of more fragmented products that remain undetected by the $NO_3^-$-TOF-CIMS. For the oxidation of naphthalene, we find increasing OOM yields under high OH exposure. Multi-generation OH oxidation may occur more efficiently for precursors with two aromatic rings than the monocyclic precursors, thereby forming more of the detectable OOMs by the $NO_3^-$-TOF-

CIMS even under high OH exposure. The parameters in the yield calculation are associated with various kinds of uncertainties. We used a uniform calibration factor to determine the OOM concentrations which may underestimate the sensitivity of large molecules. Given an average transmission efficiency of ~0.5 for 450-650 Th as suggested by Tian et al. (2023), the underestimations on OOM yields (about 0.01-0.03%) are minor. The main bias in yields is perhaps associated with $k_{loss}$ which is most sensitive to the condensation sink (Palm et al., 2016). It is worth noting that the yield calculation herein assumes that

the production rate is equal to the loss rate of OOMs, which may represent only the lower limit of the actual yield if particles are still growing when exiting the OFR.

**3.2 Formation of $C_x$ and $C_{2x}$ OOMs**

For all precursors, the concentrations of closed-shell $C_x$ products are much greater than those of open-shell products (i.e., mainly $RO_2$) (Figure S3). Under low-$NO_x$ conditions, the closed-shell $C_x$ products are formed primarily from the $RO_2 + HO_2$

pathways rather than the reaction of $RO_2 + RO_2$. This is due to the much higher $HO_2$ concentrations (0.9-2.4 ppb) than the $RO_2$ concentrations (0.1-2.2 ppt) (Table 1), which hints that the formation of closed-shell $C_x$ products is limited by the $RO_2$ availability. Indeed, good correlations between closed-shell $C_x$ products and $RO_2$ are observed (Figure 3). The slopes are generally lower for double-ring precursors than for monocyclic precursors, indicating a steric hindrance of double rings on the $C_x$ product formation.

For a given precursor of $C_xH_y$, further auto-oxidation of BPRs ($C_xH_{y+1}O_5$) may lead to the formation of $C_x$ products with hydrogen numbers of $y$ or $y+2$, whereas multi-generation OH reactions (i.e., a second or third OH attack) would lead to the formation of $C_x$ products with hydrogen numbers of $y$-2 (H abstraction) or $> y+2$ (OH addition) (Garmash et al., 2020; Cheng et al., 2021b). Figure 4 shows the signal fractions of closed-shell $C_x$ products that have different hydrogen numbers at different OH exposure. The signal fractions of products with $y$-2 hydrogen numbers increase with increasing OH exposure, suggesting

that more H abstraction may have occurred at higher OH exposure. Consistently, the signal fractions of $C_x$ products with $> y+2$ hydrogen numbers decrease as the OH exposure increases, suggesting less OH addition at higher OH exposure. H

abstraction has been found to be involved in the formation of the majority of monoterpene OOMs (Shen et al., 2022). Moreover, the signal fractions of products with $y$-2 and $> y$+2 hydrogen numbers are relatively higher in naphthalene- and 1-methylnaphthalene-derived OOMs than in other types of OOMs, highlighting the importance of multi-generation OH oxidation for double-ring aromatic precursors. Additionally, for naphthalene and 1-methylnaphthalene, both of the products with $y$+4 (15.9-27.8%) and $y$+6 (3.4-10.9%) hydrogen numbers have high signal fractions, whereas for monocyclic aromatic OOMs, the products with $y$+4 (9.7-37.9%) hydrogen numbers are the main $> y$+2 products (Tables S2-S7 and Figure S4). The presence of abundant $C_x$ products with hydrogen numbers of $y$+6 from double-ring precursors suggests a significant occurrence of third OH addition on early-generation products that still possess a high degree of unsaturation.

The concentrations of $C_{2x}$ products are several times lower than those of $C_x$ products (Figure S5). Their total concentrations vary by OH exposures but show different trends for different precursors. For benzene and 1-methylnaphthalene, the $C_{2x}$ concentrations increase for higher OH exposures, whereas for the other four precursors, the $C_{2x}$ concentrations decrease as the OH exposure increases. This highlights a large difference in the $RO_2$ distribution and their functional groups for different precursors, calling for further investigations. The concentrations of $C_{2x}$ products correlate well with the square of the $RO_2$ concentrations except for m-xylene (Figure 5). The slopes are lower for naphthalene than for monocyclic precursors, suggesting again significant steric effects of double rings on the formation of $C_{2x}$ products (Tomaz et al., 2021). In addition, the slopes are greater for methyl substituted precursors than for non-substituted ones.

Table 2 lists the constrained rate constants for benzene- and naphthalene-derived dimeric products with reasonable fittings ($r >$ 0.5). Our estimated rate coefficients for $C_{2x}$ products from the self- and cross-reactions of $RO_2$ radicals span a range of one order of magnitude (i.e., 0.9-8.5×10$^{-10}$ cm$^3$ molecule$^{-1}$ s$^{-1}$). The rates are similar to those reported for α-pinene + OH/O$_3$ and 1,3,5-trimethylbenzene + O$_3$/OH (i.e., 0.1-8.7×10$^{-10}$ cm$^3$ molecule$^{-1}$ s$^{-1}$) (Berndt et al., 2018a; Berndt et al., 2018b; Molteni et al., 2019). The estimated rate constants are 7.0×10$^{-10}$ cm$^3$ molecules$^{-1}$ s$^{-1}$ for $C_6H_7O_8$ + $C_6H_7O_9 \rightarrow C_{12}H_{14}O_{15}$ and 0.9×10$^{-10}$ cm$^3$ molecules$^{-1}$ s$^{-1}$ for $C_{10}H_9O_8$ + $C_{10}H_9O_8 \rightarrow C_{20}H_{18}O_{14}$, suggesting that the accretion reaction rate is sensitive to precursor structures and $RO_2$ functional groups. These $RO_2$ radicals could be formed from BPRs ($C_xH_{y+1}O_5$), thus the lower rate constant for $C_{10}H_9O_8$ + $C_{10}H_9O_8 \rightarrow C_{20}H_{18}O_{14}$ may again be explained by the steric influence of double rings.

### 3.3 OOM Contribution to SOA

Figure 6 shows the average elemental ratios of SOA and detected gaseous OOMs as well as the estimated contributions of the detected OOMs to the SOA mass for the experiments having AMS and SMPS particle measurements. For benzene oxidation, greater O:C ratios and lower H:C ratios are observed for SOA produced at higher OH exposures, which can be explained by typical functionalization process (Kroll et al., 2011). For toluene oxidation, the O:C ratios of SOA increase and then decrease as OH exposures increase while the H:C ratios show the opposite, indicating significant fragmentation at high OH exposures. The lowest OH exposure in the OFR (> 0.9 days atmospheric equivalent photochemical age) are greater than chamber conditions. Correspondingly, the O:C ratios at lowest OH exposure for both types of SOA are greater, and the H:C ratios are

lower compared to the reported values in previous chamber studies (Nakao et al., 2013; Chhabra et al., 2010) (Figure 6a).

Unlike benzene and toluene SOA, the H:C ratios for naphthalene SOA increase as OH exposures increase, and the O:C ratios at lowest OH exposure are lower than the reported ratios in chamber study (Chhabra et al., 2010), suggesting more difference in the oxidation conditions between the OFR and chamber studies for naphthalene. Our results agree with the reported O:C ratios in another OFR study (i.e., 0.86-1.40 for toluene SOA and 0.48-1.59 for naphthalene SOA) (Lambe et al., 2011). The detected OOMs produced from naphthalene have lower O:C and lower H:C ratios than those from benzene and toluene (Figure

6b). When the OH exposure increases, the average O:C ratios of the detected OOMs increase and the H:C ratios decrease (Figure S6). The increase of O:C ratios of OOMs is more significant for the oxidation of naphthalene, whereas the decrease of H:C ratios of OOMs is more evident for the oxidation of benzene and toluene. The much broader ranges of O:C and H:C for each type of SOA than those for detected OOMs highlight the important contributions of undetected products to SOA. Notably, the O:C ratios of naphthalene-derived OOMs are much lower than those of benzene-derived OOMs, but the naphthalene-

derived OOMs contain more low-volatility fractions than those of benzene-derived OOMs as shown in the volatility distribution (Figure S7).

The measured SOA mass concentrations are 4.3-22.2 μg m$^{-3}$ at the exit of OFR. The particle wall loss correction has much greater uncertainties than for chamber seeded experiments because that the particle size distributions can only be measured at the exit of the OFR. We therefore did not calculate the SOA yields in this study. The contributions of OOMs with $C^* > 0.01$

μg m$^{-3}$ to SOA may be evaluated by assuming instantaneous gas-particle equilibrium. On the basis of partition theory and the vapor pressure of OOMs (Pankow, 1994), the estimated contributions are about 0.3-18.3% for the oxidation of aromatics. For OOMs with $C^*$ of $\leq 0.01$ μg m$^{-3}$, their contributions to SOA are usually estimated by calculating the net condensation flux. We can only obtain the net condensation flux at the exit of OFR, which should be greater than the flux inside the OFR considering the particle growth in the OFR. Multiplying this flux with the mean residence time of the OFR, these low-volatility

species would explain 1.9-14.1% of the aromatic SOA mass. Together, the identified OOMs by NO$_3^-$-TOF-CIMS may contribute to 3.9-10.3%, 9.9-30.0%, and 6.7-18.4% of the measured SOA mass for the oxidation of benzene, toluene, and naphthalene, respectively, under our low NO$_x$ OFR conditions (Figure 6c). A recent field study suggests ~50-70% of aromatic OOM (also detected by NO$_3^-$-TOF-CIMS) contributions to SOA in urban environments in China (Nie et al., 2022), highlighting the potential importance of these OOMs to SOA. As shown in Figure S7, most of the identified OOMs are in the low-volatility

range (i.e., ULVOC, ELVOC, LVOC). The volatility distribution of OOMs from naphthalene has a greater ULVOC fraction than those of benzene, which is consistent with the greater estimated contributions of naphthalene-derived OOMs to the SOA mass. Field measurements in urban Shanghai suggest that heavy aromatic VOCs are important SOA precursors (Tian et al., 2022). The results herein provide laboratory evidence that heavy (e.g., double-ring) aromatic VOCs might be more important than light (i.e., monocyclic) aromatic ones in initial particle growth during SOA formation.

The estimation of OOM contributions to the SOA mass has large uncertainties. The main calculation uncertainty is associated with the estimation of OOM volatilities, which could be ± 1 bin different (i.e., 1 or 2 orders of magnitude in $C^*$) or more

(Donahue et al., 2011). This may affect significantly the calculation of gas-particle partitioning and net condensation flux with compensation errors among species. On the other hand, the OFR experiments may not mimic fully the ambient conditions. High OH exposures with short residence time may lead to fast $RO_2 + HO_2$ reactions that limit some $RO_2$ auto-oxidation. Precursors are mixed in real atmosphere, which may lead to different $RO_2$ chemistry and product distributions compared to single precursor systems in this study (Chen et al., 2022). Measurement uncertainties are relatively minor and not subject to significant systematic bias. The underestimation of large molecules (450-650 Th) in $NO_3^-$-TOF-CIMS analysis by a factor of 2 only affects the OOM contribution by ~0.1%. The quantification uncertainties of VOCs and SOA are usually less than 30% (Huang et al., 2019; Zheng et al., 2021). In ambient applications, the calculation of OOMs to SOA needs to isolate the observed SOA growth to OOM production, which introduces additional uncertainties related to air mass dilution and transport.

## 4 Conclusions

The formation of OOMs from the photooxidation of benzene, toluene, m-xylene, 1,3,5-trimethylbenzene, naphthalene, and 1-methylnaphthalene was investigated under low-$NO_x$ conditions with a wide range of OH exposure. The detected OOM formulae show diverse oxygen and hydrogen numbers that need both of multi-step auto-oxidation and multi-generation OH oxidation to explain, highlighting similar reaction rates of the two mechanisms under atmospheric relevant VOC concentrations. Multi-generation OH oxidation appears to be more prominent at higher OH exposure for the oxidation of double-ring precursors than for the oxidation of single-ring aromatic precursors. The product distributions demonstrate significant steric effects of the methyl substitution and the double-ring structure on the OOM formation. We show that (1) the formation of fragmented products, especially unsaturated small oxygenates, are less efficient for precursors with more methyl groups, and (2) the steric effects of double rings affect the formation of both of monomeric and dimeric products. More studies are needed for developing quantitative understanding to improve the CTM representation on the OOM formation. Finally, we found more OOMs produced by the oxidation of double-ring precursors in the lower volatility range than those formed from the oxidation of traditional monocyclic precursors, highlighting the importance of heavy aromatics to SOA formation. The detectable OOMs by the $NO_3^-$-TOF-CIMS are usually highly oxygenated products and explain at maximum 30.0% of the aromatic SOA mass herein. More of the SOA mass are perhaps contributed by less oxygenated products that require other types of chemical ionization technique to explore.

*Data availability*. Data presented in this paper are available at https://doi.org/10.5281/zenodo.10047716

*Author contributions*. QC and YJL designed the study. XC, YZ, and KL conducted the experiments. XC performed the data analysis with the help of all authors. XC, QC, and YJL wrote the manuscript with inputs from all other authors.

*Competing interests*. At least one of the (co-)authors is a member of the editorial board of Atmospheric Chemistry and Physics. The peer-review process was guided by an independent editor, and the authors also have no other competing interests to declare.

*Financial support*. This work is supported by the National Natural Science Foundation of China (41875165, 42250410322, 41961134034), the special fund of State Key Joint Laboratory of Environmental Simulation and Pollution Control (23Y05ESPCP), and the 111 Center of Urban Air Pollution and Health Effects (B20009). YJL acknowledges funding support from the Science and Technology Development Fund, Macao SAR (File no. 0023/2021/A1) and a multiyear research grant (No. MYRG2022-00027-FST) from the University of Macau. The authors gratefully acknowledge Andrew Lambe, and Penglin Ye for instrument support and helpful discussion.

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

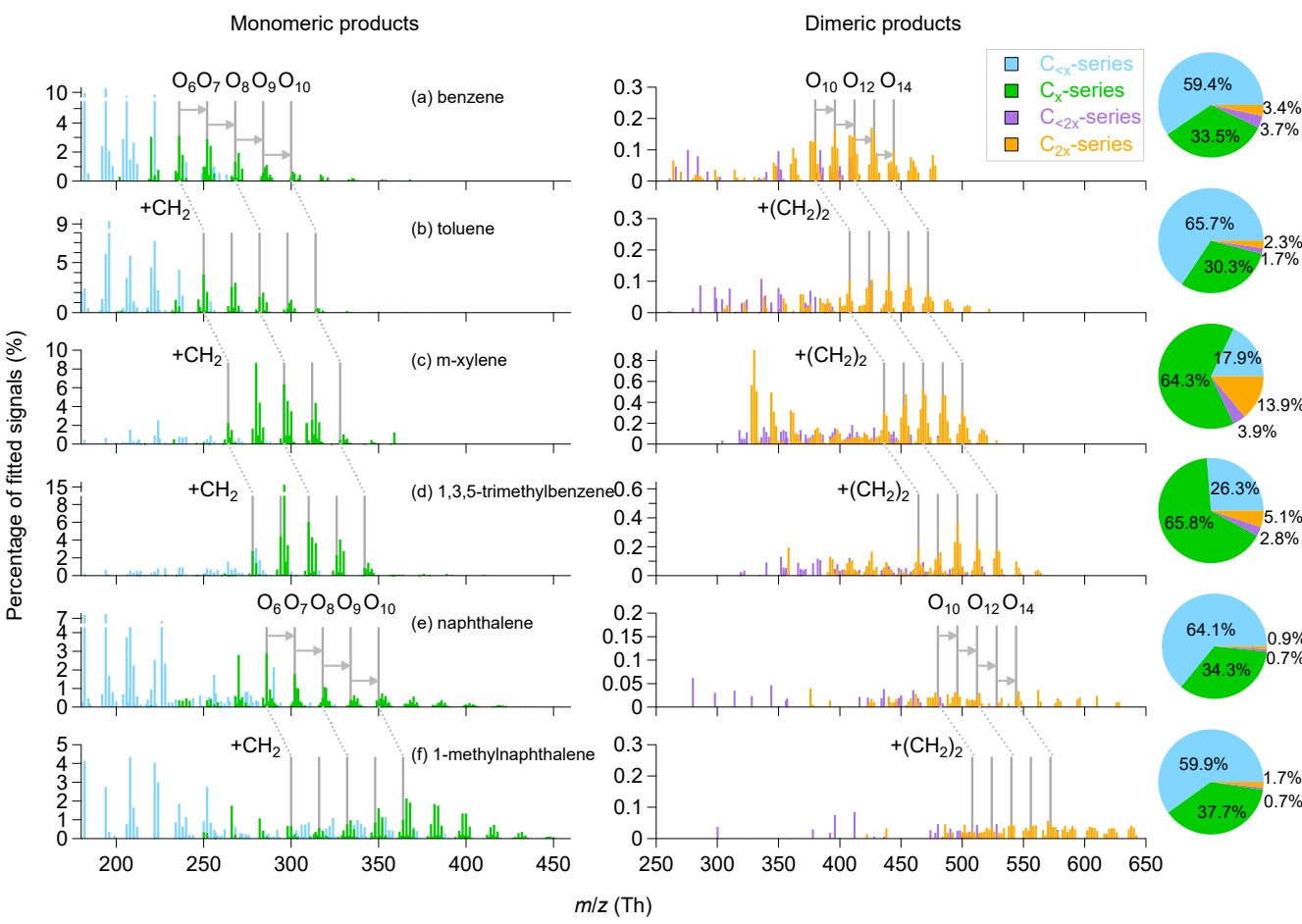

**Figure 1.** Mass spectra and fractions of different categories of gaseous OOMs observed in the oxidation of **(a)** benzene, **(b)** toluene, **(c)** m-xylene, **(d)** 1,3,5-trimethylbenzene, **(e)** naphthalene and **(f)** 1-methylnaphthalene, corresponding to experiments No. 2, 7, 12, 16, 22 and 26 in Table 1, respectively.

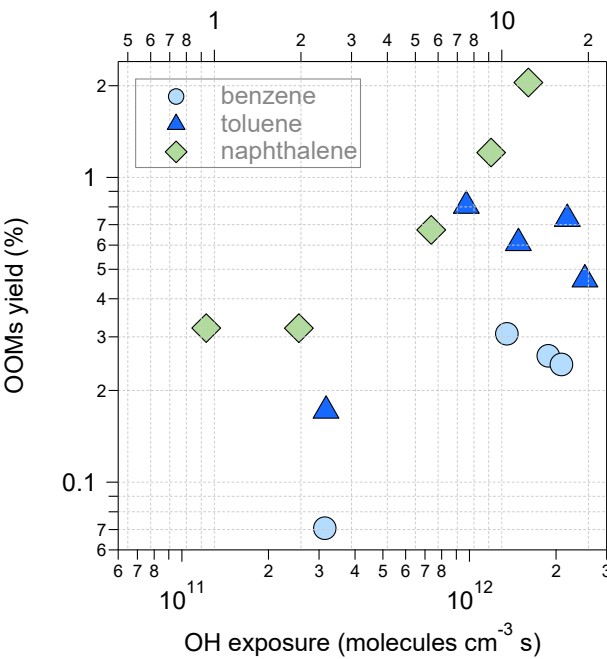

**Figure 2.** Yields of OOMs produced from photooxidation of benzene, toluene, and naphthalene in OFR as a function of OH exposure. Equivalent photochemical ages are referred to a mean OH concentration of $1.5 \times 10^6$ molecules cm$^{-3}$.

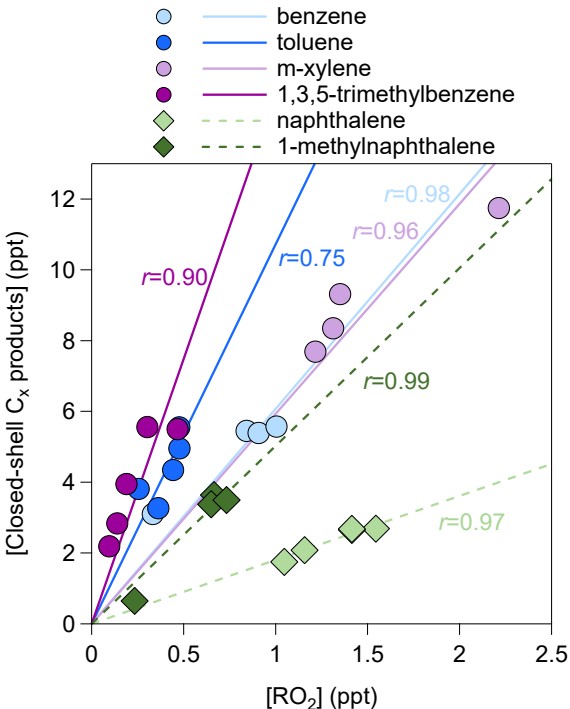

**Figure 3.** Scatter plot of the concentrations of closed-shell $C_x$ products and those of open-shell $C_x$ products ($RO_2$) for the 27 experiments listed in Table 1. The $r$ values represent the Pearson correlation coefficients.


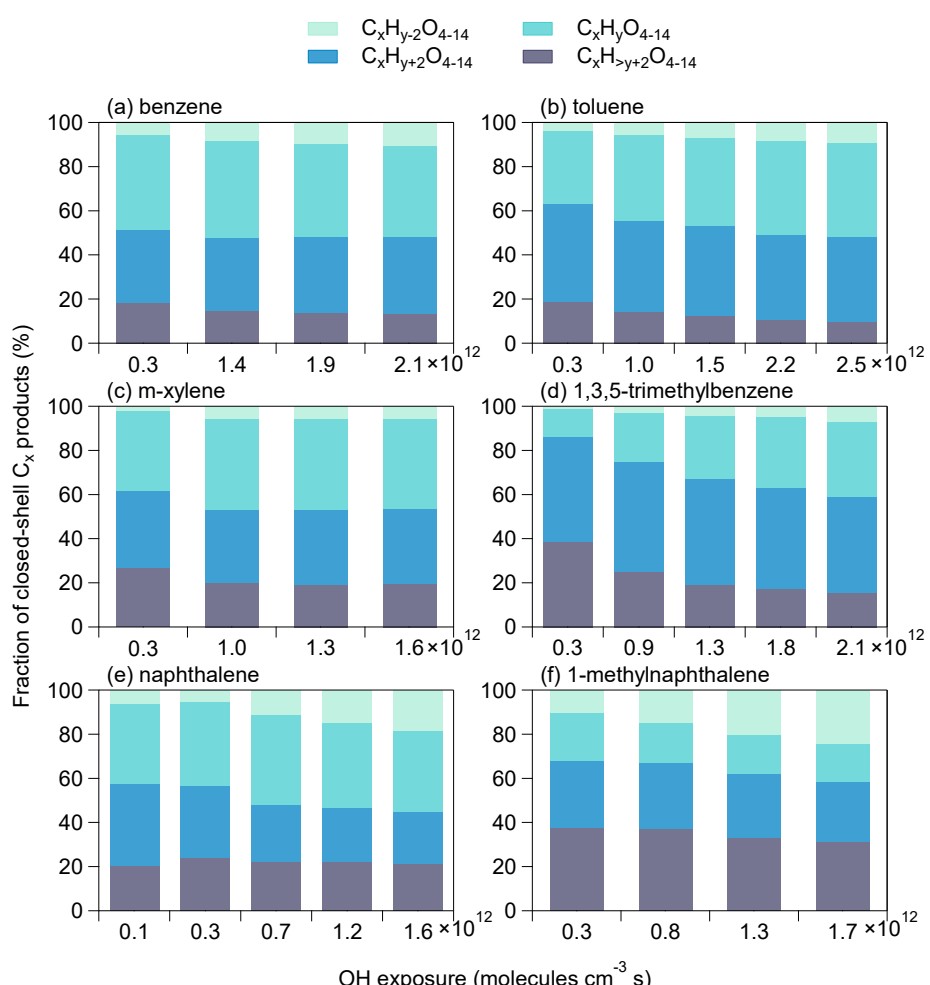

**Figure 4.** Fractions of different series of closed-shell $C_x$ products formed from the oxidation of **(a)** benzene, **(b)** toluene, **(c)** m-xylene, **(d)** 1,3,5-trimethylbenzene, **(e)** naphthalene and **(f)** 1-methylnaphthalene at different OH exposure. $x$ and $y$ represent the numbers of carbon and hydrogen atoms of the aromatic precursor ($C_xH_y$).


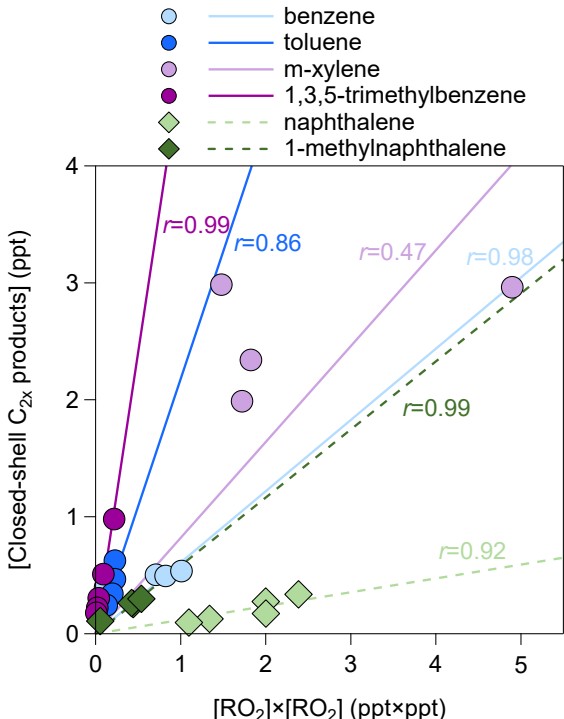

**Figure 5.** Scatter plot of the concentrations of closed-shell $C_{2x}$ products and those of $RO_2 \times RO_2$ for the 27 experiments listed in Table 1. The $r$ values represent the Pearson correlation coefficients.

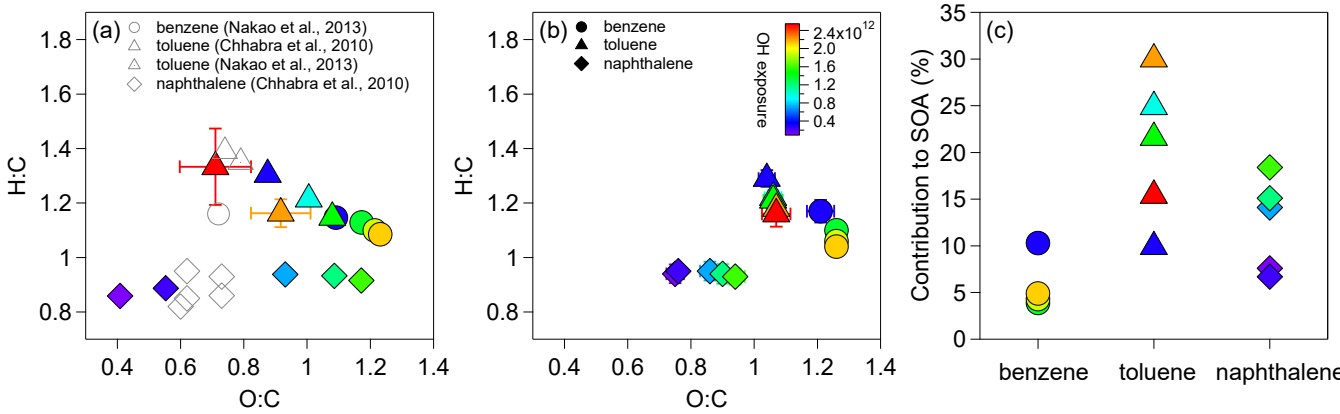


**Figure 6. (a)** The averaged H:C and O:C ratios of the resulted SOA for the oxidation of benzene, toluene and naphthalene. **(b)** The averaged H:C and O:C ratios of gaseous OOMs detected in these experiments. **(c)** Estimated contributions of the detected OOMs to the SOA mass. The volatilities of OOMs were calculated by using the method described in Wang et al. (2020a).

**Table 1.** Summary of experimental parameters, measured and derived quantities for aromatic ($C_xH_y$) oxidation in the OFR under low-$NO_x$ conditions.

| Exp. No | VOC species | Experimental Conditions | | | Measured Quantities | | | Derived Quantities | |
|---|---|---|---|---|---|---|---|---|---|
| | | VOC | RH | T | $C_x$ OOMs | $C_{2x}$ OOMs | SOA[a] | $OH_{exp}$ | $HO_2$ |
| | | (ppb) | (%) | (°C) | (ppt) | (ppt) | (µg m$^{-3}$) | (molec cm$^{-3}$ s) | (ppb) |
| 1[#] | benzene | 110 | 23.6 | 25.0 | 3.4 | 0.4 | 5.1 | $3.1 \times 10^{11}$ | 1.5 |
| 2[#] | | | 24.3 | 24.4 | 6.3 | 0.5 | 22.2 | $1.4 \times 10^{12}$ | 2.3 |
| 3[#] | | | 24.0 | 25.0 | 6.3 | 0.5 | 17.4 | $1.9 \times 10^{12}$ | 2.4 |
| 4[#] | | | 23.4 | 25.6 | 6.6 | 0.5 | 13.7 | $2.1 \times 10^{12}$ | 2.4 |
| 5[#] | toluene | 50 | 29.4 | 23.4 | 4.1 | 0.6 | 4.7 | $3.2 \times 10^{11}$ | 1.5 |
| 6[#] | | | 29.0 | 23.7 | 6.0 | 0.6 | 14.3 | $9.8 \times 10^{11}$ | 2.2 |
| 7[#] | | | 28.3 | 24.3 | 5.4 | 0.5 | 8.4 | $1.5 \times 10^{12}$ | 2.3 |
| 8[#] | | | 27.5 | 25.4 | 4.8 | 0.3 | 4.3 | $2.2 \times 10^{12}$ | 2.3 |
| 9[#] | | | 26.8 | 26.4 | 3.6 | 0.2 | 4.5 | $2.5 \times 10^{12}$ | 2.2 |
| 10 | m-xylene | 38 | 26.0 | 23.4 | 8.9 | 3.0 | n.a. | $2.8 \times 10^{11}$ | 1.5 |
| 11 | | | 27.0 | 23.1 | 14.0 | 3.0 | | $9.5 \times 10^{11}$ | 2.2 |
| 12 | | | 24.5 | 24.7 | 10.7 | 2.3 | | $1.3 \times 10^{12}$ | 2.4 |
| 13 | | | 24.1 | 24.3 | 9.7 | 2.0 | | $1.6 \times 10^{12}$ | 2.4 |
| 14 | 1,3,5-trimethylbenzene | 17 | 26.1 | 22.1 | 6.0 | 1.0 | n.a. | $2.7 \times 10^{11}$ | 1.4 |
| 15 | | | 26.2 | 23.0 | 5.9 | 0.5 | | $8.5 \times 10^{11}$ | 2.2 |
| 16 | | | 25.4 | 23.8 | 4.1 | 0.3 | | $1.3 \times 10^{12}$ | 2.3 |
| 17 | | | 24.6 | 24.8 | 3.0 | 0.2 | | $1.8 \times 10^{12}$ | 2.4 |
| 18 | | | 23.7 | 25.8 | 2.3 | 0.2 | | $2.1 \times 10^{12}$ | 2.4 |
| 19[#] | naphthalene | 12 | 28.7 | 19.5 | 4.2 | 0.3 | 7.4 | $1.2 \times 10^{11}$ | 0.9 |
| 20[#] | | | 29.4 | 19.2 | 4.1 | 0.3 | 7.5 | $2.6 \times 10^{11}$ | 1.3 |
| 21[#] | | | 28.2 | 20.0 | 4.1 | 0.2 | 6.3 | $7.4 \times 10^{11}$ | 2.0 |
| 22[#] | | | 27.4 | 20.6 | 3.2 | 0.1 | 5.5 | $1.2 \times 10^{12}$ | 2.2 |
| 23[#] | | | 26.6 | 21.5 | 2.8 | 0.1 | 4.7 | $1.6 \times 10^{12}$ | 2.3 |
| 24 | 1-methylnaphthalene | 17 | 25.6 | 23.3 | 0.9 | 0.1 | n.a. | $2.8 \times 10^{11}$ | 1.5 |
| 25 | | | 26.3 | 23.1 | 4.3 | 0.3 | | $8.5 \times 10^{11}$ | 2.2 |
| 26 | | | 25.8 | 23.5 | 4.0 | 0.3 | | $1.3 \times 10^{12}$ | 2.3 |
| 27 | | | 25.0 | 24.1 | 4.2 | 0.3 | | $1.7 \times 10^{12}$ | 2.4 |

[#] Experiments that have LTOF-SP-AMS measurements.
[a] Not corrected by particle wall losses.

**Table 2.** Rate coefficients $k$ for benzene- and naphthalene-derived dimer, and comparison with data from other studies. R squares ($R^2$) are shown to verify the fitting results.

| Dimer | $R^2$ | $RO_2 + R'O_2$ | $k$ ($10^{-10}$ cm$^3$ molecule$^{-1}$ s$^{-1}$) |
|---|---|---|---|
| $C_{12}H_{14}O_{15}$ | 0.31 | $C_6H_5O_9 + C_6H_9O_8$ | 3.0 |
| | | $C_6H_7O_7 + C_6H_7O_{10}$ | 6.4 |
| | | $C_6H_7O_8 + C_6H_7O_9$ | 7.0 |
| $C_{20}H_{18}O_{14}$ | 0.45 | $C_{10}H_5O_5 + C_{10}H_{13}O_{11}$ | 6.0 |
| | | $C_{10}H_5O_6 + C_{10}H_{13}O_{10}$ | 8.5 |
| | | $C_{10}H_5O_7 + C_{10}H_{13}O_9$ | 7.3 |
| | | $C_{10}H_7O_5 + C_{10}H_{11}O_{11}$ | 1.7 |
| | | $C_{10}H_7O_6 + C_{10}H_{11}O_{10}$ | 2.4 |
| | | $C_{10}H_7O_9 + C_{10}H_{11}O_7$ | 5.2 |
| | | $C_{10}H_7O_{11} + C_{10}H_{11}O_5$ | 3.9 |
| | | $C_{10}H_9O_6 + C_{10}H_9O_{10}$ | 1.2 |
| | | $C_{10}H_9O_8 + C_{10}H_9O_8$ | 0.9 |
| $C_{18}H_{26}O_8$[a] | / | $C_9H_{13}O_5 + C_9H_{13}O_5$ | 1.4-2.5 |
| $C_{20}H_{30}O_6$[b] | / | $C_{10}H_{15}O_4 + C_{10}H_{15}O_4$ | 0.1 |
| $C_{20}H_{34}O_8$[b] | / | $C_{10}H_{17}O_5 + C_{10}H_{17}O_5$ | 0.4 |
| $C_{20}H_{30}O_{18}$[b] | / | $C_{10}H_{15}O_{10} + C_{10}H_{15}O_{10}$ | 0.5 |
| $C_{20}H_{32}O_{14}$[b] | / | $C_{10}H_{17}O_7 + C_{10}H_{15}O_9$ | 0.8 |
| $C_{20}H_{30}O_{14}$[c] | / | $C_{10}H_{15}O_8 + C_{10}H_{15}O_8$ | 3.2 |
| $C_{20}H_{30}O_{15}$[c] | / | $C_{10}H_{15}O_{10} + C_{10}H_{15}O_7$ | 8.7 |
| | | $C_{10}H_{15}O_9 + C_{10}H_{15}O_8$ | 6.6 |
| $C_{20}H_{30}O_{16}$[c] | / | $C_{10}H_{15}O_{10} + C_{10}H_{15}O_8$ | 2.3 |
| $C_{20}H_{30}O_{17}$[c] | / | $C_{10}H_{15}O_{11} + C_{10}H_{15}O_8$ | 4.4 |
| | | $C_{10}H_{15}O_{10} + C_{10}H_{15}O_9$ | 1.8 |
| $C_{20}H_{30}O_{18}$[c] | / | $C_{10}H_{15}O_{12} + C_{10}H_{15}O_8$ | 1.6 |
| | | $C_{10}H_{15}O_{10} + C_{10}H_{15}O_{10}$ | 0.8 |

[a] from experiments with 1,3,5-trimethylbenzene addition (Berndt et al., 2018b)
[b] from experiments with α-pinene addition (Berndt et al., 2018a)
[c] from experiments with α-pinene addition (Molteni et al., 2019)