# Peer review of "Oxygenated organic molecules produced by low-NOx photooxidation of aromatic compounds: contributions to secondary organic aerosol and steric hindrance"

_EGUsphere, 2023_

## Author Comment (AC1)

**Response to reviews**

Reviewer comments are in **bold**. Author responses are in plain text labeled with [R]. Line numbers in the responses correspond to those in the revised manuscript (the version with all changes accepted). Modifications to the manuscript are in *italics*.

**Editor**

**The authors will still have to show more clearly during peer review how they arrive at their statement "the OOMs identified by the NO3−-TOF-CIMS perhaps consist of 3-11% of the SOA mass" (abstract, l. 27) and why this is only "perhaps" the case.**

[R0] Thanks for the suggestions. We have revised the parts in Sect. 2.4, Lines 293-319, and Supplement Sect. S1 to describe in more detail about the assumptions, parameters, and uncertainties of the calculation of OOM contributions to SOA and also updated the relevant conclusion in the abstract (Line 25).

**Reviewer #1**

**Cheng et al. present in this study a systemic investigation of the oxidation of multiple aromatic VOCs using an oxidation flow reactor. To start with, the authors performed detailed analyses on the oxidation products measured by a nitrate CIMS, by which they showed the importance of both multi-generation oxidation and autoxidation in producing OOMs and the significant influence of steric hindrance in intra-molecular H-shift and dimer formation. The authors further estimated the accretion reaction rate constants between RO2 radicals, which are consistent with the values in previous literature. In the end, the authors estimated the contribution of OOMs to SOA via condensation and equilibrium partitioning, which appeared to be much lower than the value estimated from ambient measurement in a recent study (Nie et al., 2022). In this regard, the inconsistency points out the substantially incomplete understanding of the role of OOMs in SOA formation.**

**In general, I think this topic is of high importance, and this manuscript is well-structured and easy to follow. However, I do have some concerns that need to be addressed before it can be accepted for publication.**

[R0] We thank the reviewer for the valuable feedback. Detailed responses to the comments are given below.

**Major concerns:**

**I appreciate that the authors mentioned the weak representativeness of OFR to atmospheric conditions (Line 265). However, I worry that this message is not clear enough and can be easily overlooked. In Line 264, the authors say "Large uncertainties remain in the estimation", which is just handwaving. The authors should explicitly list possible sources of uncertainties, which help navigate the knowledge gap for future research.**

[R1] We thank the reviewer for the suggestion. We have expanded the discussion of the uncertainties in Lines 309-319 as follows: "The estimation of OOM contributions to the SOA mass has large uncertainties. The main calculation uncertainty is associated with the estimation

of OOM volatilities, which could be ± 1 bin different (i.e., 1 or 2 orders of magnitude in $C^*$) or more (Donahue et al., 2011). This may affect significantly the calculation of gas-particle partitioning and net condensation flux with compensation errors among species. On the other hand, the OFR experiments may not mimic fully the ambient conditions. High OH exposures with short residence time may lead to fast $RO_2 + HO_2$ reactions that limit some $RO_2$ auto-oxidation. Precursors are mixed in real atmosphere, which may lead to different $RO_2$ chemistry and product distributions compared to single precursor systems in this study (Chen et al., 2022). Measurement uncertainties are relatively minor and not subject to significant systematic bias. The underestimation of large molecules (450-650 Th) in $NO_3^-$-TOF-CIMS analysis by a factor of 2 only affects the OOM contribution by ~0.1%. The quantification uncertainties of VOCs and SOA are usually less than 30% (Huang et al., 2019; Zheng et al., 2021). In ambient applications, the calculation of OOMs to SOA needs to isolate the observed SOA growth to OOM production, which introduces additional uncertainties related to air mass dilution and transport.".

**Some specific comments are listed below:**
**(Line 65-66) The OH and HO2 concentration is disproportionally high in the experiment, which affects the competition among different reaction channels of OOM formation. First, the RO2 termination reaction is dominated by RO2+HO2 reactions; Second and more importantly, the fast RO2+HO2 reaction (due to high HO2 concentration) could lead to a very short lifetime of RO2 radicals that limits the RO2 autoxidation. This should be clearly discussed (at least mentioned) in Sect. 2.1.**

[R2] We thank the reviewer for the suggestion. We have added the discussion in Lines 84-87 as follows: "Because the $HO_2$ concentrations were 2-18 times greater than the OH concentrations in the OFR, we expect that the $RO_2$ termination was dominated by $RO_2 + HO_2$ reactions in most of the OFR experiments herein. The $RO_2$ isomerization rate coefficients are highly structure dependent, ranging from ~$10^{-3}$ to a few tens of s$^{-1}$ for aromatic $RO_2$ (Wang et al., 2017; Praske et al., 2018). Therefore, the fast $RO_2 + HO_2$ reactions may limit some $RO_2$ auto-oxidation in our experiments."

**(Line 103-104) Besides a general calibration factor, do the authors consider the mass-dependence transmission efficiency of the instrument (Heinritzi et al., 2016)? A steep transmission curve can significantly affect the signal strength, affecting the concentration estimation (SOA calculation) and the determination of accretion reaction rate constants.**

[R3] We agree with the reviewer that Heinritzi et al. (2016) showed a steep increase in their transmission curve from ~62 to 550 Th by a factor of ~5 by using perfluorinated acids as calibrants. However, Tian et al. (2023) and Zheng et al. (2023) did similar calibrations and showed much smooth transmission curves that peak at 114 or 269 Th by an enhance factor of 1.01 or 1.22. We found similar calibration factors for $H_2SO_4$ and 4NPh (201 Th) for our instrument (see the Table below). Besides, others report similar calibration factors for the range of 97-426 Th. We therefore decide to use a uniform calibration factor in this study. This may underestimate the concentrations of large molecules by a factor of 2 given the average

transmission efficiency of ~0.5 for 450-650 Th suggested by Tian et al. (2023). Because the OOM signal fractions of 450-650 Th are about 0.4-4.9% herein, such an underestimation may lead to an overall increase of 0.1-0.2% in the estimated OOM contribution values to SOA. For the determination of accretion reaction rate constants, a concentration underestimation by a factor of 2 would change the rate coefficient by up to a factor of 2. We have clarified all these in Lines 106-111 and 315-317.

| Compounds | $m/z$ (Th) | Calibration factor ($\times 10^{10}$ molecules cm$^{-3}$) | References |
|---|---|---|---|
| sulfuric acid | 97.07 | 1.3
 1.1
 0.9 | Jokinen et al. (2012); Ehn et al. (2014)
 Kürten et al. (2012)
 Cheng et al. (2021) |
| 4-nitrophenol (4NPh) | 201.11 | 1.23 | Cheng et al. (2021) |
| perfluoroheptanoic acid (PFHA) | 426.06 | 0.96 | Ehn et al. (2014) |

**(Line 108-110) Is the steady state a good assumption for OFR conditions? The stable concentration at each individual experimental condition could also be interpreted as that the chemical reactions are stable in the OFR, so the formation and loss of OOMs at a constant residence time yields a stable concentration (not necessarily at the steady state). Can the authors provide data or calculations to support this assumption, or is there any previous literature discussing this?**

[R4] We agree with the reviewer that stable concentrations do not necessarily indicate steady state conditions. Study indicated fast production of α-pinene derived OOM dimers ($\sim 10^{-10}$ cm$^3$ molecules$^{-1}$ s$^{-1}$) (Molteni et al., 2019). Theoretical studies also showed that the ROOR′ formation can occur through the fast association rate around $(0.5-2)\times 10^{-10}$ cm$^3$ molecules$^{-1}$ s$^{-1}$ (Hasan et al., 2020; Perakyla et al., 2023). In addition, we observed similar signal trends for the dimers (e.g., $C_{14}H_{18}O_{14}$) that we were constraining with their parent RO$_2$ (e.g., $C_7H_9O_{10}$) (see below). This supports fast production of these dimers.

Furthermore, the PAMchem model simulations suggest minor changes of the ROOR′ concentrations when the reaction rate constant for RO$_2$+R′O$_2$→ROOR′ reaches above $5\times 10^{-11}$ cm$^3$ molecules$^{-1}$ s$^{-1}$ (Lambe et al., 2017). The rate coefficients that we can constrained are at the $10^{-10}$ cm$^3$ molecules$^{-1}$ s$^{-1}$ level. We therefore think the steady-state assumption may stand for these dimers (which does not work for the dimers produced by slow production). To clarify, we revised the text in Lines 126-132.

[Figure]

**Also, there is evidence that ROOR' could further react with OH, forming different ROOR" (Wang et al., 2020). Did the authors consider this reaction as a loss/source term of ROOR' when deriving the $k_{R,i}$?**

[R5] Yes, we have considered this reaction in the loss term. To clarify, we revised the text in Sect. S1 of SI as follows, "we followed the assumption used in Palm et al. (2016) (1) all OOMs including ROOR' may react with OH for up to five generations, (2) the rate constant for reaction with OH ($k_{OH}$) is $1.0 \times 10^{-11}$ cm$^3$ molecule$^{-1}$ s$^{-1}$.".

**(Line 126-127, and the corresponding text in SI) The parameterization by Mohr et al., (2019) are more suitable for OOMs from monoterpene oxidation, which contains several hydroperoxyl groups, which may not apply to OOMs from monoterpene oxidation. In fact, Wang et al., (2020) showed that this is not suitable for naphthalene products and provided a new parameterization. I suggest adopting the one by Wang et al., (2020). Also, it seems that the temperature-dependence of dHvap (in eq. S3) is different than the one used in e.g., Stolzenburg et al., (2018). The authors need to reference this equation. These will affect the volatility distribution of OOMs and the estimation of the contribution to SOA.**

[R6] We thank the reviewer for the suggestion. We have adopted the parameterizations used by Wang et al. (2020) in the calculation and have revised in Lines 137-141, Lines 292-300, and Section S1 for the discussion. This parameterization leads to higher estimated contributions of detected OOM to SOA but won't change our conclusions. We have also updated Fig. 6c (see below) for the updated contribution values.

[Figure]

For $\Delta H_{vap}$, Equation S3 is taken from Epstein et al. (2010). In this equation, we applied a $\Delta\theta_{10}$ value of 5.7 kJ mole$^{-1}$ instead of 11 kJ mole$^{-1}$ according to Donahue et al. (2011). We have added these references in Line 141 and Sect. S1 of SI to clarify the calculation of $\Delta H_{vap}$.

**(Line 186-188) As the authors stated and consistent with tradition knowledge, BPRs are the central radicals. Assuming that dimers are formed from RO2+RO2 -> ROOR + O2, the least oxidized C2x dimer would be CxHyO8. Given this assumption, the observation of abundant O4-7 dimers is interesting. Can the authors speculate the formation pathway?**

[R7] We think the $C_{2x}$ products with 4-7 oxygen numbers might be formed from $RO_2$ radicals that are less oxygenated than BPRs. For example, $C_8H_9O_2$ and $C_8H_{11}O_3$ are expected $RO_2$ radicals produced by xylene oxidation (Birdsall et al., 2011). Two $C_8H_{11}O_3$ radicals may form an $O_4$ dimer. $C_8H_9O_2$ and BPRs may form $O_5$ dimers. $C_8H_{11}O_3$ and BPRs may form $O_6$ dimers. To clarify this, we have added more discussion about this in Lines 198-201.

**(Line 240) I am confused about this. Isn't that C12H14O15 can form via different combinations of RO2 radicals (as in Table 2). Then how can the $k_{R,I}$ be derived with only one exclusive combination?**

[R8] Yes, each dimer can form via different combinations of $RO_2$ radicals. As described in Sect. 2.3, each combination has a rate constant of $k_{R,i}$. In this study, only 5-8 dimers have sufficient combinations of above-detection-limit signals of possible $RO_2$ and $R'O_2$ to solve $k_{R,i}$. Only the $k_{R,i}$ values that are greater than zero but less than $9\times10^{-10}$ cm$^3$ molecule$^{-1}$ s$^{-1}$ with correlation coefficients ($r$) of $> 0.5$ are considered as reasonable rate constants and are listed in Table 2. We compared the constrained $k_{R,i}$ in our study to the literature values for a specific combination. To clarify this, we have revised the text in Lines 266-269.

**Minor comment**
**(Line 105) Cheng et al., 2021a is missing from the reference list.**

[R9] We have revised the reference list.

References:
Birdsall, A. W., et al., Comprehensive NO-dependent study of the products of the oxidation of atmospherically relevant aromatic compounds. *J. Phys. Chem. A* **2011,** *115*, (21), 5397-5407.
Chen, T., et al., Secondary organic aerosol formation from mixed volatile organic compounds: Effect of RO2 chemistry and precursor concentration. *npj Climate and Atmospheric Science* **2022,** *5*, (1), 95.
Cheng, X., et al., Secondary production of gaseous nitrated phenols in polluted urban environments. *Environ. Sci. Technol.* **2021,** *55*, (8), 4410-4419.
Donahue, N. M., et al., A two-dimensional volatility basis set: 1. organic-aerosol mixing thermodynamics. *Atmos. Chem. Phys.* **2011,** *11*, (7), 3303-3318.
Ehn, M., et al., A large source of low-volatility secondary organic aerosol. *Nature* **2014,** *506*, (7489), 476-479.
Epstein, S. A., et al., A Semiempirical Correlation between Enthalpy of Vaporization and

Saturation Concentration for Organic Aerosol. *Environ. Sci. Technol.* **2010,** *44*, (2), 743-748.

Hasan, G., et al., Comparing Reaction Routes for 3(RO・・・OR′) Intermediates Formed in Peroxy Radical Self- and Cross-Reactions. *The Journal of Physical Chemistry A* **2020,** *124*, (40), 8305-8320.

Heinritzi, M., et al., Characterization of the mass-dependent transmission efficiency of a CIMS. *Atmos. Meas. Tech.* **2016,** *9*, (4), 1449-1460.

Huang, G., et al., Potentially important contribution of gas-phase oxidation of naphthalene and methylnaphthalene to secondary organic aerosol during haze events in Beijing. *Environ. Sci. Technol.* **2019,** *53*, (3), 1235-1244.

Jokinen, T., et al., Atmospheric sulphuric acid and neutral cluster measurements using CI-APi-TOF. *Atmos. Chem. Phys.* **2012,** *12*, (9), 4117-4125.

Kürten, A., et al., Calibration of a chemical ionization mass spectrometer for the measurement of gaseous sulfuric acid. *J. Phys. Chem. A* **2012,** *116*, (24), 6375-6386.

Lambe, A., et al., Controlled nitric oxide production via O(1-D) + N2O reactions for use in oxidation flow reactor studies. *Atmos. Meas. Tech.* **2017,** *10*, (6), 2283-2298.

Molteni, U., et al., Formation of highly oxygenated organic molecules from α-pinene ozonolysis: chemical characteristics, mechanism, and kinetic model development. *ACS Earth Space Chem.* **2019,** *3*, (5), 873-883.

Palm, B. B., et al., In situ secondary organic aerosol formation from ambient pine forest air using an oxidation flow reactor. *Atmos. Chem. Phys.* **2016,** *16*, (5), 2943-2970.

Perakyla, O., et al., Large Gas-Phase Source of Esters and Other Accretion Products in the Atmosphere. *J. Am. Chem. Soc.* **2023**.

Praske, E., et al., Atmospheric autoxidation is increasingly important in urban and suburban North America. *Proc. Natl. Acad. Sci. U. S. A.* **2018,** *115*, (1), 64-69.

Tian, L., et al., Enigma of Urban Gaseous Oxygenated Organic Molecules: Precursor Type, Role of NOx, and Degree of Oxygenation. *Environ. Sci. Technol.* **2023,** *57*, (1), 64-75.

Wang, M., et al., Photo-oxidation of aromatic hydrocarbons produces low-volatility organic compounds. *Environ. Sci. Technol.* **2020,** *54*, (13), 7911-7921.

Wang, S., et al., Formation of highly oxidized radicals and multifunctional products from the atmospheric oxidation of alkylbenzenes. *Environ. Sci. Technol.* **2017,** *51*, (15), 8442-8449.

Zheng, P., et al., Molecular Characterization of Oxygenated Organic Molecules and Their Dominating Roles in Particle Growth in Hong Kong. *Environ. Sci. Technol.* **2023**.

Zheng, Y., et al., Precursors and Pathways Leading to Enhanced Secondary Organic Aerosol Formation during Severe Haze Episodes. *Environ. Sci. Technol.* **2021,** *55*, (23), 15680-15693.

**Reviewer #2**

**The paper presents a set of experiments conducted in an oxidation flow reactor (OFR) targeted at speciating and quantifying oxygenated organic molecules (OOM) in a reaction of aromatic VOCs with OH. Six common aromatic VOCs and 4 to 5 OH exposures per precursor were sampled. Oxidation products are detected by a nitrate-scheme chemical ionization mass spectrometry. The study is limited to low-NOx regime and high precursor conditions adding to previous experimental studies a wider range of conditions, precursors as well as extending the research question to the contribution of the detected species to secondary organic aerosol (SOA). Unfortunately, quantification of molar yields and SOA contribution is only available for 3 out of 5 VOCs. The manuscript is well written and results are adequately discussed. However, there are few places in the paper, where I would like to see some clarifications regarding the uncertainties, methods and discussion. I present my comments below.**

[R0] We thank the reviewer for the valuable feedback. Detailed responses to the comments are given below.

**Major comments.**

**1. The paper lacks overall discussion on the limitations of the study in representing real atmosphere or in providing quantitative results.**

**a) The experiments are conducted at high VOC and OH concentrations, higher than some previous studies. How could this affect the observed OOM composition and yields?**

[R1] We agree with the reviewer that the VOC and OH concentrations are higher than some previous chamber studies (Molteni et al., 2019; Garmash et al., 2020). In our previous study, we have compared the experimental conditions and major OOMs for the photooxidation of benzene with previous low-$NO_x$ studies (Table S4 in the SI of Cheng et al. (2021)) and discussed the influence of experimental conditions on the OOM composition and yields. Overall, the observed OOM formulae are quite similar among studies. A longer residence time can promote multi-step auto-oxidation, and indeed our results show more abundant $O_7$ and $O_9$ products from benzene oxidation. Ambient environments have much lower OH concentrations but longer residence times depending on meteorological conditions, which may lead to more oxygenated products from multi-steps of auto-oxidation. Moreover, the differences in $HO_2$ and $RO_2$ concentrations among different studies that remain unclear might affect the extent of auto-oxidation. To clarify, we have added some discussion to highlight the potential influence of experimental conditions and referred to our previous paper for more discussion in Lines 84-87 and 312-315.

**b) What are the uncertainties associated with calculating OOM molar yield as well as the contribution of OOM condensation to SOA? For instance, molar yield calculation includes correction due to losses, and that correction is likely large. KOHloss is an approximation of the loss due to the reaction with OH. How sensitive is the yield to the uncertainty in KOHloss or other loss parameters? Authors reference their previous study (Cheng et al. 2021) as well**

as Palm et al. (2016). However, those studies used a constant kOH for a saturated C10 molecule to approximate KOHloss. Is that value still valid for compounds like naphthalene, products of which likely contain double bonds? The authors also point to Cheng et al. (2021) to see how loss to aerosols is calculated. However, it is unclear if the same diffusion volume is being used for all VOCs or not. Authors should clarify specific methodology in the current manuscript and discuss uncertainties/biases associated with the choices made. Could you provide short comment on how CIMS was calibrated and what is the uncertainty of the calibration factor? Were any corrections applied to the calibration factor for lower oxygenated OOMs that are not detected at collision limit in nitrate-CIMS? If no correction was done, is there a possibility for bias in interpretation of OOM contribution to SOA and/or molar yields?**

[R2] The $k_{OH}$ values for OOMs are largely unknown. So we followed the assumption used in previous studies to comparison purpose. Palm et al. (2016) showed insensitive OOM fates to $k_{OH}$ and the number of reactions with OH in their sensitivity analysis. With the assumed $k_{OH}$ and maximum reaction steps, we found that the majority of the OOM loss in the OFR came from the condensation sink. In the calculation of condensation sink, we used specific diffusion volume that are calculated on the basis of the mean atomic ratios of the observed OOMs and ring structures for each type of VOC under different OH exposures. These are now clarified in Sect. S1.

Moreover, as described in [R3] for Review #1, we calibrated the nitrate CIMS with sulfuric acid and 4-nitrophenol (4NPh). We found similar calibration factors for $H_2SO_4$ and 4NPh (201 Th) for our instrument. Others report similar calibration factors for the range of 97-426 Th. We therefore decide to use a uniform calibration factor herein. This may underestimate the concentrations of large molecules by a factor of 2 given the average transmission efficiency of ~0.5 for 450-650 Th suggested by Tian et al. (2023). Because the OOM signal fractions of 450-650 Th are about 0.4-4.9% herein, such an underestimation may lead to an overall increase of 0.1-0.2% in the estimated OOM contribution values to SOA and an increase of 0.01-0.03% in the reported total OOM yields, both of which are minor biases. We have added the uncertainty and potential bias discussion in Lines 106-111, Lines 309-317 and Lines 224-230.

**2. As this paper presents quantitative results motivated by improving the available datasets, it would be reasonable that the authors would deposit the data (at least the data used for making figures) in a persistent repository.**

[R3] Yes, the data are now available in a public repository as described in the Data availability section.

**3. Some aspects of SOA production in current experiments remain unclear.**
**a) It would be useful to see some description if SOA was produced in nucleation rather than aerosol seed experiments. It would help to understand the system if authors could provide details on how much of aerosol mass was produced and how SOA yields compared to**

**previous nucleation studies. Also, to which sizes did the particles grow (at least in terms of understanding detection by AMS)?**

[R4] We thank the reviewer for the suggestion. The SOA concentrations ranged from 4.3-22.2 µg m$^{-3}$ and the particles grew to around 70 nm (volume mode size in mobility diameter). We have added the SOA mass concentrations in Table 1 and clarified that "the OFR experiments were conducted without injection of seed particles" in Line 75.

The OFR conditions are unlike chamber conditions that mimic better ambient environment. Lambe et al. (2015) showed that SOA yields obtained in the Aerodyne OFR were lower than those obtained in chamber studies at similar OH exposure levels. We obtained SOA yields of about 0.05-0.16 (corrected for wall loss) for benzene, toluene, and naphthalene oxidation, which are indeed lower than the yields of 0.15-0.49 for toluene oxidation under low NO$_x$ and non-seeded chamber conditions reported by Xu et al. (2015). However, for nucleation-type OFR experiments, the particle wall loss correction has much greater uncertainties than for chamber seeded experiments because that (1) the SMPS size distributions can only be measured at the exit of the OFR and (2) particles smaller than 50 nm have significantly higher wall loss coefficients even in chamber conditions (Park et al., 2001). We therefore think the SOA yield results measured herein are not very meaningful. To explain, we added some description in Lines 291-293.

**b) HOM (a subset of OOM) are known to be most important for SOA growth at lower SOA concentrations (Ehn et al. 2014), exactly what one would expect in suburban or downwind low-NOx conditions. How the amount of SOA produced in current experiments could potentially bias the results (OOM to SOA contribution)? Some discussion on this should be included in the text.**

[R5] We have added the measured SOA concentrations in Table 1 (i.e., about 4.3 to 22.2 µg m$^{-3}$), which are in the atmospherically relevant range for polluted suburban environments in China. We agree with the reviewer that the estimation of the OOM contribution to SOA has large uncertainties. The main purpose of conducting such estimation is to compare with ambient findings using the same estimation method to explore the importance of these highly oxygenated products to SOA. As responded in [R1, R3, R6] for Reviewer #1, we have added more discussion and revised some of the calculations and potential bias for clarification.

**c) If this study looked at nucleation experiments, is it possible to derive OOM importance at different particle growth stages (provided short residence times in OFR would allow for this)? This could help to further illustrate the relative importance of different VOCs in early particle growth.**

[R6] We thank the reviewer for the suggestion. However, all the OFR experiments were conducted under ~95 s residence time. Unlike chamber studies, we cannot track the nucleation events and examine the early particle growth stage. Particles have already grown to big sizes that AMS was able to detect at the exit.

**Minor comments.**

**Author present OOM molar yields. How do new results compare to the previous study by the same author (Cheng et al. 2021) in terms of yield values and OH exposures?**

[R7] We refitted the spectra for the oxidation of benzene and toluene in this study and thus the molar yields are slightly different from what we reported in the paper by Cheng et al. (2021). To clarify, we add a description in Lines 103-106 and a table in SI: "The oxidation of benzene and toluene have been discussed in our previous study for a fitted mass-to-charge ratio (*m/z*) range of 150 to 450 Thomson (Th) (Cheng et al., 2021). In this study, we refitted the spectra to up to 650 Th, resulting in slightly different molar yields (Table S1)".

**From Figures S1 and S4, it is clear that the concentration of dimers for some compounds decreases at increasing irradiance in OFR (or increasing OH). It is also not consistent among different precursors, e.g. naphthalene vs 1-methylnaphthalene. Could some additional discussion be presented in C2x section (section 3.2)?**

[R8] We thank the reviewer for the suggestion. We add additional discussion in Lines 254-258 as: "Their total concentrations vary by OH exposures but show different trends for different precursors. For benzene and 1-methylnaphthalene, the $C_{2x}$ concentrations increase for higher OH exposures, whereas for the other four precursors, the $C_{2x}$ concentrations decrease as the OH exposure increases. This highlights a large difference in the $RO_2$ distribution and their functional groups for different precursors, calling for further investigations."

**Figures 3,5: when printed, the colors for toluene and benzene appear identical. Would be good to change the color as symbols are the same.**

[R9] Thanks for the suggestion. We have changed the color settings for benzene and toluene in Figures 2, 3 and 5.

**Figure 6 a,b: same axes limits would be useful.**

[R10] We have set the axes to the same scale in Figure 6a,b.

**Line 47. Some logical transition between sentences on multi-generation oxidation and dimer formation is needed.**

[R11] We added "Additionally" for the transition.

**Line 53: 'subtraction' – did you mean 'abstraction'?**

[R12] Yes, we have changed it to "abstraction".

**Lines 151-152: 'suggesting significant hydrogen loss in the dimer formation' – what do authors mean by this? Is it being suggested that hydrogen atoms are lost in RO2+RO2 reaction?**

[R13] We mean that losing H atoms via H abstraction occurred before accretion reactions. To clarify, we have revised the text here as "suggesting significant H abstraction process before the accretion reactions occur".

**Line 166 and 176: 'neutrals' – as both open and closed shell products are electrically neutral, it is best to use just 'closed shell' in these sentences.**

[R14] We have used "closed shell products" instead.

**Line 229: 'have high signals' – here it would be useful to provide some numbers within the text.**

[R15] We have added the signal fractions of $y+4$ and $y+6$ products for naphthalene and 1-methylnaphthalene in Line 250.

**Line 233: 'They correlated well' – from Figure 5, it seems xylene is an exception. Please clarify this in the text.**

[R16] We have clarified this in Lines 258-259: "The concentrations of $C_{2x}$ products correlate well with the square of the $RO_2$ concentrations except for m-xylene"

**Line 234: 'The slopes are lower for naphthalene and 1-methylnaphthalene'. It seems that the slope for naphthalene is identical to that of benzene. Would be good to clarify the text.**

[R17] We have clarified this in Lines 259-261: "The slopes are lower for naphthalene than for monocyclic precursors, suggesting again significant steric effects of double rings on the formation of $C_{2x}$ products (Tomaz et al., 2021). In addition, the slopes are greater for methyl substituted precursors than for non-substituted ones.".

**Line 248: 'most of the O:C ratios are much greater while the H:C ratios are lower'. From Figure 6a, I can see that the H:C for naphthalene is on the same order as previous studies, while O:C is somewhere lower or higher. For toluene, O:C and H:C ratios are similar at highest OH exposure, while for benzene, H:C ratios are similar. This is in contrast with the text. Authors should be more specific when interpreting the comparison.**

[R18] We thank the reviewer for the suggestion. We have revised the discussion in Lines 272-281 as follows, "For benzene oxidation, greater O:C ratios and lower H:C ratios are observed for SOA produced at higher OH exposures, which can be explained by typical functionalization process (Kroll et al., 2011). For toluene oxidation, the O:C ratios of SOA increase and then decrease as OH exposures increase while the H:C ratios show the opposite, indicating significant fragmentation at high OH exposures. The lowest OH exposure in the OFR (> 0.9 days atmospheric equivalent photochemical age) are greater than chamber conditions. Correspondingly, the O:C ratios at lowest OH exposure for both types of SOA are greater, and the H:C ratios are lower compared to the reported values in previous chamber studies (Nakao et al., 2013; Chhabra et al.,

2010) (Figure 6a). Unlike benzene and toluene SOA, the H:C ratios for naphthalene SOA increase as OH exposures increase, and the O:C ratios at lowest OH exposure are lower than the reported ratios in chamber study (Chhabra et al., 2010), suggesting more difference in the oxidation conditions between the OFR and chamber studies for naphthalene."

References:

Cheng, X., et al., Highly oxygenated organic molecules produced by the oxidation of benzene and toluene in a wide range of OH exposure and NOx conditions. *Atmos. Chem. Phys.* **2021,** *21*, (15), 12005-12019.

Chhabra, P. S., et al., Elemental analysis of chamber organic aerosol using an aerodyne high-resolution aerosol mass spectrometer. *Atmos. Chem. Phys.* **2010,** *10*, (9), 4111-4131.

Garmash, O., et al., Multi-generation OH oxidation as a source for highly oxygenated organic molecules from aromatics. *Atmos. Chem. Phys.* **2020,** *20*, (1), 515-537.

Kroll, J. H., et al., Carbon oxidation state as a metric for describing the chemistry of atmospheric organic aerosol. *Nature Chem.* **2011,** *3*, 133.

Lambe, A. T., et al., Effect of oxidant concentration, exposure time, and seed particles on secondary organic aerosol chemical composition and yield. *Atmos. Chem. Phys.* **2015,** *15*, (6), 3063-3075.

Molteni, U., et al., Formation of highly oxygenated organic molecules from α-pinene ozonolysis: chemical characteristics, mechanism, and kinetic model development. *ACS Earth Space Chem.* **2019,** *3*, (5), 873-883.

Nakao, S., et al., Density and elemental ratios of secondary organic aerosol: application of a density prediction method. *Atmos. Environ.* **2013,** *68*, 273-277.

Palm, B. B., et al., In situ secondary organic aerosol formation from ambient pine forest air using an oxidation flow reactor. *Atmos. Chem. Phys.* **2016,** *16*, (5), 2943-2970.

Park, S. H., et al., Wall loss rate of polydispersed aerosols. *Aerosol Sci. Technol.* **2001,** *35*, (3), 710-717.

Tian, L., et al., Enigma of Urban Gaseous Oxygenated Organic Molecules: Precursor Type, Role of NOx, and Degree of Oxygenation. *Environ. Sci. Technol.* **2023,** *57*, (1), 64-75.

Tomaz, S., et al., Structures and reactivity of peroxy radicals and dimeric products revealed by online tandem mass spectrometry. *Nat. Commun.* **2021,** *12*, (1), 300.

Xu, J. L., et al., Simulated impact of NOx on SOA formation from oxidation of toluene and m-xylene. *Atmos. Environ.* **2015,** *101*, 217-225.

---

## Author Response (AR2)

**Response to reviews**

Reviewer comments are in **bold**. Author responses are in plain text labeled with [R]. Line numbers in the responses correspond to those in the revised manuscript (the version with all changes accepted).

**Editor**

**I would like to ask the authors to expand on their discussion of RO2 fate. In l. 86 of the revised manuscript with track changes, they write: "Because the HO2 concentrations were 2-18 times greater than the OH concentrations in the OFR, we expect that the RO2 termination was dominated by RO2 + HO2 reactions in most of the OFR experiments herein." Please give more details on what leads you to that expectation. Were the RO2 fates calculated? If yes, please show these results. If not, I recommend performing a simple calculation of RO2 fates under the conditions in the OFR.**

[R0] Thanks for the suggestion. The modeled $HO_2$ concentrations were typically 10 times greater than the OH concentrations in most of our OFR experiments. The rate coefficient for $RO_2$ + $HO_2$ was typically an order of magnitude smaller than that for $RO_2$ + OH (i.e., ~$1.5\times10^{-11}$ vs. ~$1.0\times10^{-10}$ $cm^3$ $molecules^{-1}$ $s^{-1}$) (Peng et al., 2020, and references therein). Therefore, we expect that $RO_2$ + $HO_2$ plays the dominant role in $RO_2$ fate in most of the OFR experiments herein. We have modified text in Lines 84-87.

References:

Peng, Z., et al., Radical chemistry in oxidation flow reactors for atmospheric chemistry research. *Chem. Soc. Rev.* **2020**, *49*, (9), 2570-2616.

**Reviewer #1**

**I appreciate that the authors expand the discussion of CIMS calibration. However, it needs to be pointed out that the transmission of instruments vary from one to another, depending on the tuning of each instrument. Therefore, it is not fair to justify the "good tuning" by the results of previous studies. In Cheng et al., 2021, the authors did show that the transmission appeared to be stable between sulfuric acid (97 and 160 m/z) and 4NPh (201 m/z). But the authors should be aware of the closeness between these signals, which does not guarantee a flat transmission on a wider range of mass-to-charge. With this said, it is perhaps difficult for the authors to evaluate the transmission during their experiments a while ago. Therefore, I think it necessary to mention the potential uncertainty introduced by transmission in the manuscript.**

[R0] We thank the reviewer for the suggestion. We have clarified the potential uncertainty introduced by transmission efficiency in the main text as shown in Lines 108-114.

---

## Author Response (AR3)

**Response to editor**

We thank the editor for the constructive comments that help improve the manuscript. We provide below the responses to those comments. Editor comments are in **bold**. Author responses are in plain text labeled with [R]. Line numbers in the responses correspond to those in the revised manuscript with track changes. Modifications to the manuscript are in *italics*.

**In l. 84 of the revised manuscript with track changes, the authors now write:**

**"The HO2 concentrations were 2-18 times (over 10 times in most of our OFR experiments) greater than the OH concentrations in the OFR. The rate coefficient for RO2 + HO2 was typically an order of magnitude smaller than that for RO2 + OH (i.e., ~1.5×10-11 vs. ~1.0×10-10 cm3 molecules-1 s-1) (Peng and Jimenez, 2020, and references therein). We therefore expect that the RO2 termination was dominated by RO2 + HO2 reactions in most of the OFR experiments herein."**

**My comments to this new paragraph are:**

**1. "The rate coefficient for RO2 + HO2 was typically an order of magnitude smaller than that for RO2 + OH" - Use of "was" is deceptive here as it suggests that these coefficients were measured or otherwise derived in this study.**

**2. The authors report that HO2 concentrations are about an order of magnitude higher than OH concentrations (evidence?), but RO2 + OH rate coefficients are about an order of magnitude higher than RO2 + HO2 rate coefficients, so from the reported number I would expect that both channels are of roughly similar importance. Note that Peng and Jimenez write in the mentioned reference that "Thus, RO2 + OH may play an important role in RO2 fate in OH OFR, when RO2 + HO2 is a major RO2 loss pathway (i.e. at low NO) and the HO2-to-OH ratio is close to or lower than 10. (Peng et al. 2019)". Hence, from my point of view, the conclusion "We therefore expect that the RO2 termination was dominated by RO2 + HO2 reactions in most of the OFR experiments" is not supported by the presented evidence and reference.**

**3. Calculation results of the PAMchem model are still not provided as requested.**

[R0] We sincerely thank the editor for the further comment and suggestion. We have revised the text and added a figure for the PAMchem model results in SI (shown below). We agree with the editor that both $RO_2 + HO_2$ and $RO_2 + OH$ channels were probably of similar importance. Here is the revised text (Lines 84-89):

"The modeled $HO_2$-to-OH concentration ratio is about 2-18, while the rate constant for $RO_2 + HO_2$ is typically an order of magnitude smaller than that for $RO_2 + OH$ (Peng et al., 2020, and references therein). Figure S1 shows the calculated contributions of various pathways to the $RO_2$

loss in our experiments for a given set of rate constants. Both of the $RO_2 + HO_2$ and $RO_2 + OH$ channels are important for $RO_2$ loss under conditions of our experiments considering the uncertainties of the estimations and the variations of reaction constants."

[Figure]

Figure S1. The relative contributions of $RO_2$ loss channels to the $RO_2$ fate calculated by using the $RO_2$ fate estimator (https://sites.google.com/site/pamwiki/estimation-equations) with the measured $RO_2$ concentrations, the PAMchem-model-derived OH and $HO_2$ concentrations, and the rate constants of $1.5 \times 10^{-11}$ cm$^3$ molecules$^{-1}$ s$^{-1}$ for $RO_2 + HO_2$, $1.0 \times 10^{-10}$ cm$^3$ molecules$^{-1}$ s$^{-1}$ for $RO_2 + OH$, 0.01 s$^{-1}$ for $RO_2$ isomerization, and $1.0 \times 10^{-13}$ cm$^3$ molecules$^{-1}$ s$^{-1}$ for primary $RO_2 +$

RO$_2$ reactions for all our experiments (Ziemann et al., 2012; Yan et al., 2016; Praske et al., 2018; Orlando et al., 2012).

Reference:

Orlando, J. J., et al., Laboratory studies of organic peroxy radical chemistry: an overview with emphasis on recent issues of atmospheric significance. *Chem. Soc. Rev.* **2012,** *41*, (19), 6294-6317.

Peng, Z., et al., Radical chemistry in oxidation flow reactors for atmospheric chemistry research. *Chem. Soc. Rev.* **2020,** *49*, (9), 2570-2616.

Praske, E., et al., Atmospheric autoxidation is increasingly important in urban and suburban North America. *Proc. Natl. Acad. Sci. U. S. A.* **2018,** *115*, (1), 64-69.

Yan, C., et al., Kinetics of the Reaction of CH3O2 Radicals with OH Studied over the 292-526 K Temperature Range. *J. Phys. Chem. A* **2016,** *120*, (31), 6111-6121.

Ziemann, P. J., et al., Kinetics, products, and mechanisms of secondary organic aerosol formation. *Chem. Soc. Rev.* **2012,** *41*, (19), 6582-6605.